# Emergency department crowding and mortality in 14 Swedish emergency departments, a cohort study leveraging the Swedish Emergency Registry (SVAR)

**Björn af Ugglas** [1,2] *, **Per Lindmarker** [1,2], **Ulf Ekelund** [3], **Therese Djärv** [1,2], **Martin J. Holzmann** [1,2]

1 Department of Medicine, Solna, Karolinska Institutet, Stockholm, Sweden, 2 Theme of Emergency and Reparative Medicine, Karolinska University Hospital, Stockholm, Sweden, 3 Faculty of Medicine, Department of Clinical Sciences Lund, Emergency Medicine, Lund University, Lund, Sweden

* bjorn.af.ugglas@ki.se

## Abstract

### Objectives

There is evidence that emergency department (ED) crowding is associated with increased mortality, however large multicenter studies of high quality are scarce. In a prior study, we introduced a proxy-measure for crowding that was associated with increased mortality. The national registry SVAR enables us to study the association in a more heterogenous group of EDs with more recent data. The aim is to investigate the association between ED crowding and mortality.

### Methods

This was an observational cohort study including visits from 14 EDs in Sweden 2015–2019. Crowding was defined as the mean ED-census divided with expected ED-census during the work-shift that the patient arrived. The crowding exposure was categorized in three groups: low, moderate and high. Hazard ratios (HR) for mortality within 7 and 30 days were estimated with a cox proportional hazards model. The model was adjusted for age, sex, triage priority, arrival hour, weekend, arrival mode and chief complaint. Subgroup analysis by county and for admitted patients by county were performed.

### Results

2,440,392 visits from 1,142,631 unique patients were analysed. A significant association was found between crowding and 7-day mortality but not with 30-day mortality. Subgroup analysis also yielded mixed results with a clear association in only one of the three counties. The estimated HR (95% CI) for 30-day mortality for admitted patients in this county was 1.06 (1.01–1.12) in the moderate crowding category, and 1.11 (1.01–1.22) in the high category.

**Data Availability Statement:** The data used in this study contain potentially identifying or sensitive patient information and may only be shared to

researchers after acquiring an approval from the Swedish Ethical Review Authority. The process to get access to the data is to apply for ethical approval at ansokan@etikprovning.se. Detailed instructions are available at https://etikprovningsmyndigheten.se/. When the ethical approval is granted, you apply for access to the data at Uppsala Clinical Research center (UCR), that coordinate many Swedish quality registries, using this email: datauttag@ucr.uu.se More information on the process can be found at https://www.ucr.uu.se/sv/tjanster/blanketter-och-dokument. Data access is controlled by UCR and the Swedish Emergency Registry (SVAR) https://www.ucr.uu.se/svar/forforskare/for-forskare. The public authority responsible for protecting the personal data of the patients included in this registry is the Karolinska University Hospital https://www.karolinska.se/en/karolinska-university-hospital/. The authors followed the above process and did not have any special access privileges to the data that future researchers would not have.

**Funding:** The authors received no specific funding for this work.

**Competing interests:** Dr. Djärv was supported by the Stockholm County Council (clinical research appointment). Dr. Holzmann reports receiving consultancy honoraria from Actelion, Idorsia, and Pfizer. He holds research positions funded by the Swedish Heart-Lung Foundation (grant 20170804) and the ALF agreement between the Stockholm County Council and Karolinska Institutet (grant 20170686). This does not alter our adherence to PLOS ONE policies on sharing data and materials.

## Conclusions

The association between crowding and mortality may not be universal. Factors that influence the association between crowding and mortality at different EDs are still unknown but a high hospital bed occupancy, impacting admitted patients may play a role.

## Introduction

### Background

Emergency department (ED) crowding is a global challenge, and there is overwhelming evidence of negative consequences to both patients and staff [1]. Crowding is for example associated with mistakes [2], delayed interventions [3–5] and adverse events [4,6,7] together with stress [8], burnout and dissatisfaction among staff [9]. Crowding has also been shown to be associated with increased mortality [10–13]. However large multicenter studies of high quality are still scarce. In our prior study [10], including almost all ED patient visits in the Stockholm County during 2012–2016, a new proxy-measure of ED crowding was introduced and defined as the mean ED census divided by the expected ED census during a shift at the particular ED. The measure was associated with increased 30-day mortality, but this has so far not been confirmed in other studies and settings.

In a study of a University Hospital ED in Belgium there was no association between ED crowding and mortality [6], suggesting that this association is not universal. The absolute level of crowding at an ED may impact the association, and earlier studies have recognized that crowding is worse in larger ED's [14], while smaller rural ED's tend to have better performance in this perspective [15].

The Swedish national quality registry for emergency departments "Svenska Akutvårdsregistret" (SVAR) [16,17] includes recent data from 14 EDs in four different counties in Sweden. This makes it possible to study the potential association between our crowding measure and mortality in a heterogenous group of EDs.

The rate of adverse events is highest in the first 4 days [18] after an ED visit and it would be of interest to evaluate the association between ED crowding and mortality within 7 days. It is reasonable that mortality within 7 days is more closely related to the quality of ED care than the more commonly used 30-day mortality.

### Importance

The causes, consequences and solutions to crowding have been widely studied, but these issues require a system-wide approach to address [1]. A better understanding of the association between crowding and increased mortality may contribute to an improved awareness and prioritization of the crowding problem among decision-makers.

### Goals of this investigation

The aim of this study was to investigate the association between ED crowding and all-cause mortality within 7 and 30 days from the ED visit, and the potential differences between three counties in Sweden.

## Methods

### Study design and setting

This is an observational cohort study leveraging the national quality registry for EDs in Sweden, SVAR. The registry contains data from 14 EDs in four counties and includes different types of ED's ranging from large university hospitals to smaller rural EDs. Data originates from the various electronic health care (EHR) records in the hospitals and all ED visits are automatically uploaded to SVAR on a daily basis. The registry is growing and EDs were joining the registry during the study period.

### Selection of participants

All visits from patients aged 18 years or above to the 14 EDs participating in the SVAR registry were included from January 1, 2015 to December 31, 2019. The calculation of actual and predicted ED census were based on all visits. The survival analysis required more detailed data so visits with a temporary personal identification number in the EHR were excluded. These numbers are given to foreign citizens or when the identity of the patient is protected or unknown at the time of the visit. These visits were removed since all EDs did not have a working matching logic for temporary personal ID's, and since follow-up data on mortality was unreliable and difficult to find. Patients who were dead on arrival to the ED, or where any information required in the regression model was missing, were also excluded.

### Data sources and measurement

All data originated from the SVAR registry [16,17]. Patient visit information on arrival date and time, age, sex, triage priority, chief complaint, arrival mode, admission status together with the outcomes LOS and date of death (if applicable) were analyzed. From the arrival date and time, we derived the discrete variables shift, weekday/weekend and hour. The day-shift was assumed to be between 08:00–14:59, the evening-shift 15:00–22:59 and the night-shift 23:00–06:59. Weekend was defined as starting on the Friday at 17:00 and ending at the Monday at 06:59. On public holidays, the weekend was defined as starting at 17:00 the day before the holiday and ending at 06:59 the day after the holiday. Hour was defined as an integer between 0 and 23 where 0 was including arrivals between 00:00:00 and 00:59:59. Age at arrival to the ED was rounded down to full years and divided into age groups 18–39, 40–59, 60–79 and 80 or above for the descriptive tables. For the regression analysis we used the number of full years as a continuous variable. Triage priority was taken directly from the registry. Unfortunately, the Stockholm county had a different definition of priority, using the last registered priority during the ED visit instead of the first registered priority. In general, the priority is usually lowered during the visit as actions are taken to stabilize the patient and as the most dangerous diagnosis are sequentially ruled out. Chief complaint was taken directly from the registry where the complaints are standardized across all included hospitals. To limit the number of chief complaints we identified the top 25 complaints with regards to the number of deaths during the study period. All other chief complaints were lumped into the group "Other". The arrival mode was defined as "Emergency Medical Services" (EMS) if the patient arrived with ambulance or helicopter, and all other modes of arrival were defined as "Other". Admission was defined as any admission to inpatient care at the hospital of the ED, or at another hospital. Patients that died during the ED visit were also counted as admitted. Patients admitted to care at an external geriatric unit or in a nursing home was not counted as an admission. ED LOS was defined as the time from patient registration in the EHR to the time the patient physically left the ED.

## Exposure

The crowding exposure was defined as the mean hourly ED census during the shift that the exposed patient arrived, divided with the expected census for the same shift. The expected census was estimated using a separate linear model for each ED with year, weekday/weekend and hour as predictors. For example, a large ED at 4 PM during a weekday will have a much higher expected ED census than a small ED in the middle of the night during a weekend. The exposure was categorized in three categories: Reference (0–75% of observations), moderate (75%–95% of observations), and high (highest 5% of observations) [10].

A visual explanation of the definition can be found in Figs 1 and 2. The ED census was calculated through looping through the data for each hospital and hour during the study period using the arrival and discharge time to calculate the number of patients present at each hour. There were 455 visits where the length of stay (LOS) in the ED was more than 48 hours, indicating most likely an incorrect discharge time in most cases. The LOS and discharge time for these visits were set to 48 hours. Additionally, there were 25,358 visits with missing discharge dates and times, and we then assumed that the LOS was equal to the mean LOS during the study period. There were 2,863 visits where the prediction model predicted a mean ED census for the work-shift of less than 1 patient. The predicted ED census was set to 1 patient during these shifts.

## Outcome

The primary outcome was all-cause mortality within 30 days, and the secondary outcome was all-cause mortality within 7 days. The date of death was taken directly from SVAR which imports this information from the Swedish national population register.

## Study size

In order to achieve a statistical power of 90% and a certainty of 95% with an expected mortality of 1.5% we estimated that 2,224,311 visits were needed to identify a hazard ratio of 1.08 in the high category of exposure including the top 5% of visits, and 529,564 visits to identify the same relative risk in the moderate category including 20% of visits [19].

## Statistical analysis

Patient visit characteristics were presented as absolute numbers and column percentage of ED visits by crowding category and variable. We used a Cox proportional hazards model to estimate adjusted hazard ratios (HR) with 95% confidence intervals (CI). The p-value threshold for statistical significance was set at 0.05. The model was stratified by hospital, which means that the model allows for independent baseline hazards across hospitals but assume that the HR is the same for all hospitals. The regression analyses model was adjusted for age, sex, priority, arrival hour, weekend, arrival mode and chief complaint to limit the impact of potential confounding factors. The underlying time dimension in the model was calendar date so that we could avoid bias due to known or unknown seasonality effects like the flu-season or summer holidays. Follow-up started at the date of the ED visit and ended at death, or at the latest 7- or 30 days following the visit. A person could have more than one visit within a 7- or 30-day period, but to ensure that no patient contributed with risk time more than once for each date, the following visits were left-truncated. This means that the follow-up period for the subsequent visits did not start until the follow-up period of the previous visit ended. Subgroup analysis for the counties Skåne, Stockholm and Östergötland was performed for all patients, and for only admitted patients using the same methodology as in the primary analysis. Regarding

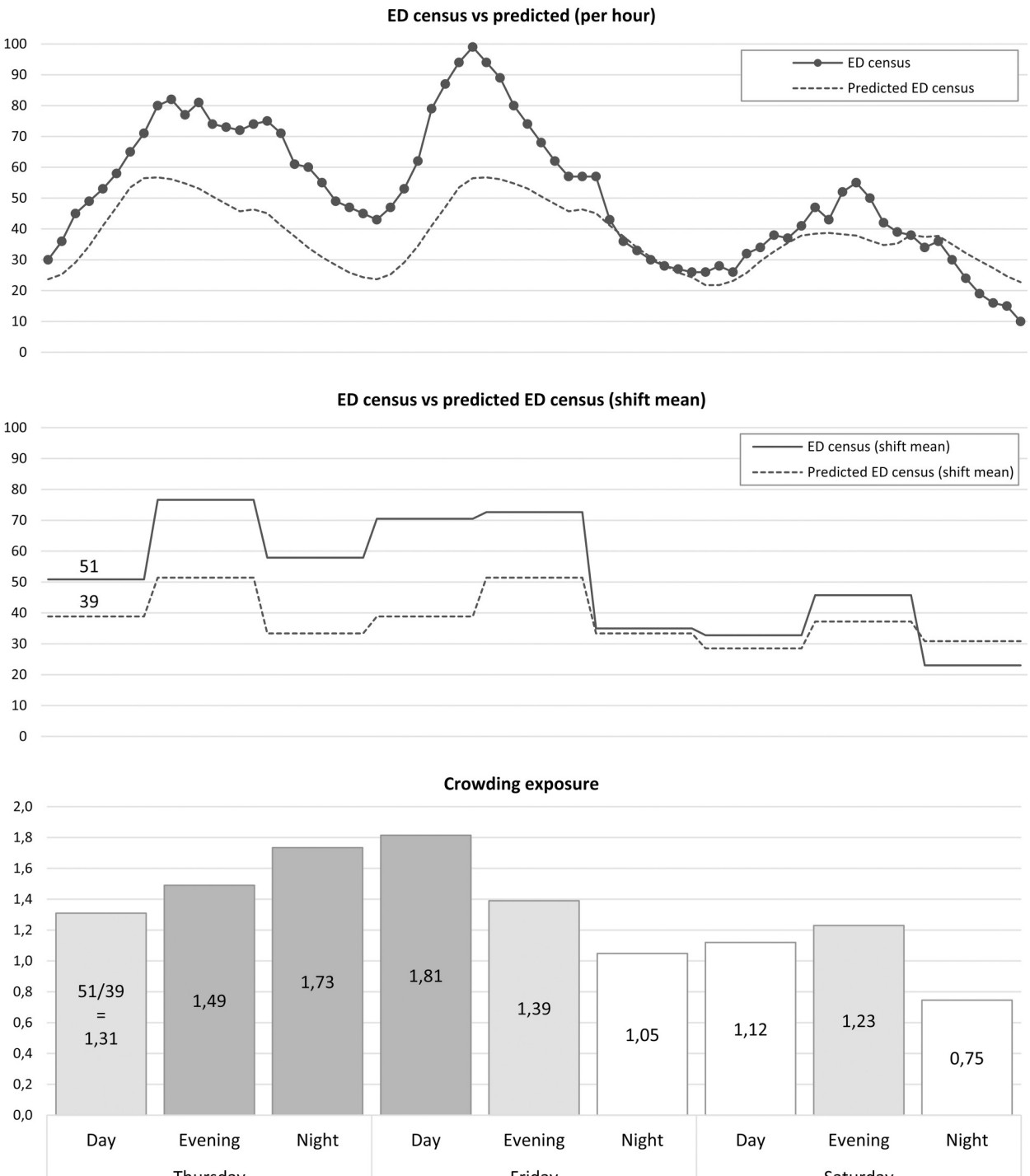

**Fig 1. Definition of crowding exposure.** The actual and predicted ED census for each specific ED and hour is calculated and one example can be seen in the top graph. From these numbers, the work-shift mean is calculated as can be seen in the middle graph. The exposure for all patients arriving during a specific shift is defined as the actual ED census for each work-shift divided with the predicted ED census for that shift, as can be seen in the bottom graph. For instance, the mean ED census during the Thursday dayshift was 51 patients. The mean expected ED census for the same shift was 39. This means that the crowding exposure for all patients arriving during the Thursday dayshift was 51/39 = 1.31.

**Distribution of exposure**

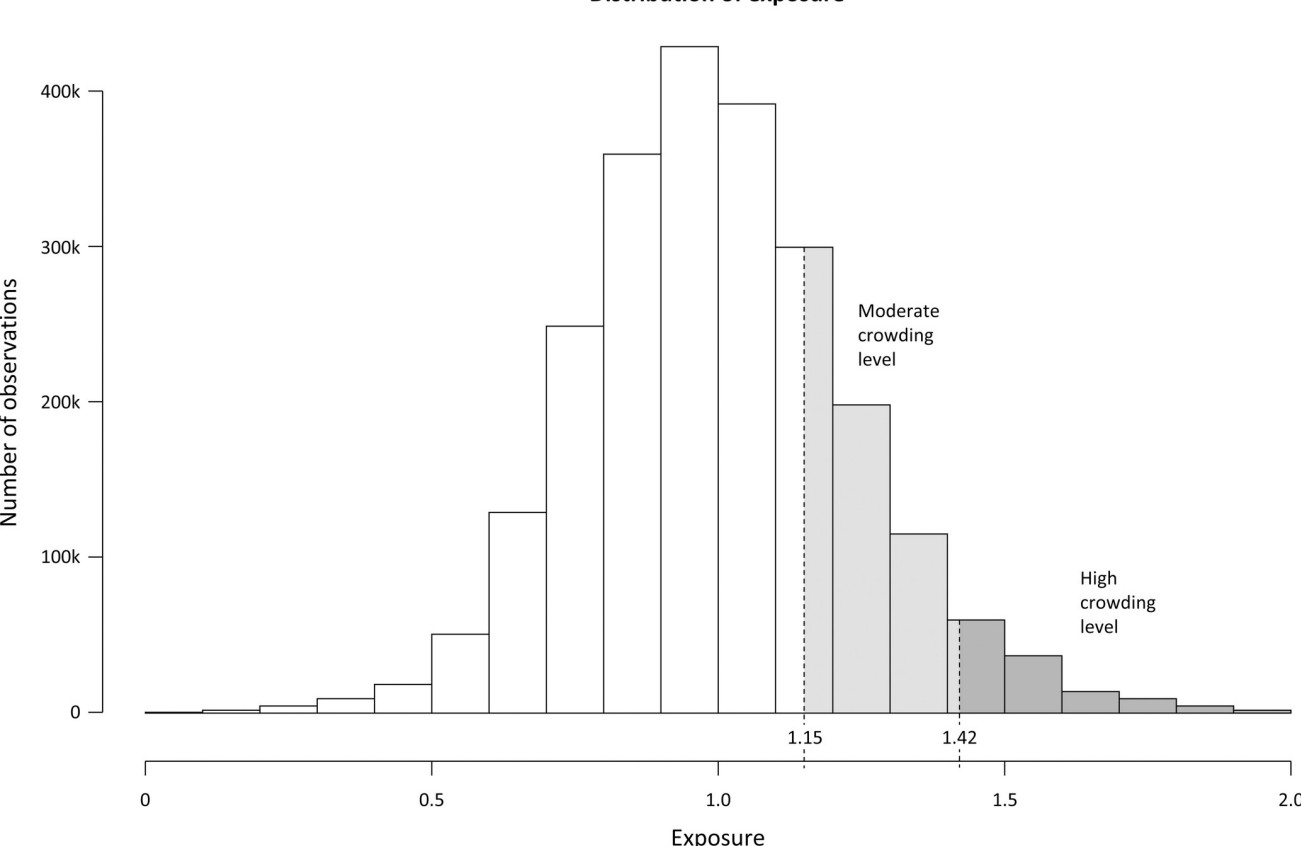

**Fig 2. Categorization of crowding exposure into crowding categories.** All patients are ranked depending on their exposure and categorized into the three crowding categories. In the primary analysis including all ED's, the threshold between the reference category and moderate category of crowding was 1.15. The threshold between the moderate and high category was 1.42.

Stockholm county we did not include visits to the Solna site at the Karolinska University Hospital after 31 March 2018 in the subgroup analysis, since this ED was transformed into a highly specialized ED with a limited scope and mandatory referral. Subgroup analysis was not performed for Örebro county since there was only one ED included with a limited number of observations. Sensitivity analysis was performed for the primary analysis including visits to all EDs. In the sensitivity analysis, the model did not include adjustment for the triage priority as the definition was different in the Stockholm county. Data management and statistical analyses were performed with R version 3.6.1 using RStudio 1.1.463.

## Ethical approval

The study was approved by the Swedish Ethical Review Authority (#2020–00120).

## Results

### Characteristics of study subjects

2,493,942 visits from 1,184,590 individual patients were extracted from SVAR. 52,363 visits were removed as they had a temporary personal identification number. 1,183 visits were discarded as patients were dead on arrival to the ED, and 4 visits were excluded due to missing information on age. Altogether, 53,550 visits (2.2%) were excluded from the original cohort

**Table 1. ED characteristics by county and hospital in primary analysis.**

| County | Hospital | ED LOS, mean (SD) | Age, median (Q1-Q3) | EMS arrival proportion, % | Admission proportion, % | 30-day mortality, proportion of visits % | 30-day mortality, incidence rate per 100 patient-years |
|--------|----------|-------------------|---------------------|---------------------------|-------------------------|------------------------------------------|--------------------------------------------------------|
| Skåne | Helsingborg | 280 (286) | 54 (34–72) | 16% | 27% | 1,7% | 26 |
| | Kristianstad | 221 (183) | 55 (34–74) | 16% | 32% | 1,7% | 24 |
| | Lund | 307 (250) | 57 (36–74) | 18% | 29% | 1,8% | 25 |
| | Malmö | 278 (211) | 56 (35–74) | 21% | 31% | 2,1% | 29 |
| | Ystad | 267 (216) | 62 (41–75) | 18% | 30% | 1,8% | 26 |
| Stockholm | Danderyd | 304 (225) | 60 (40–76) | 35% | 42% | 1,7% | 24 |
| | Huddinge | 330 (251) | 52 (34–70) | 25% | 33% | 1,4% | 19 |
| | Solna | 272 (195) | 52 (34–69) | 25% | 31% | 1,9% | 25 |
| | Södersjukhuset | 363 (262) | 55 (36–73) | 35% | 43% | 1,6% | 23 |
| | Södertälje | 222 (167) | 53 (34–72) | 23% | 26% | 0,9% | 12 |
| Örebro | Örebro | 229 (143) | 55 (34–73) | 0%[1] | 28% | 1,6% | 23 |
| Östergötland | Motala | 202 (138) | 62 (41–76) | 28% | 21% | 1,8% | 26 |
| | Linköping | 224 (142) | 53 (32–72) | 26% | 21% | 1,4% | 19 |
| | Norrköping | 213 (134) | 56 (35–73) | 32% | 25% | 1,6% | 23 |

ED: Emergency department, LOS: Length of stay, EMS: Emergency medical services.

[1] Information regarding arrival mode was not available for Örebro.

leaving 2,440,392 visits from 1,142,631 unique patients included in this study which was 97.8% of the original number of visits in the registry. ED characteristics and number of included visits are described in Tables 1 and 2.

Patient characteristics are given in Table 3. The patients' median (Q1-Q3) age for visits was 56 (36–73) years and 52% of the visits were by females. Patient characteristics were similar across the different categories of crowding with a few exceptions. The proportion of patient visits with the two highest levels of acuity were slightly more common in the high crowding category with 17.7% of visits compared to 15.3% in the lowest reference category. The

**Table 2. Number of included visits by county, hospital and year in primary analysis.**

| County | Hospital | 2015 | 2016 | 2017 | 2018 | 2019 | Total |
|--------|----------|------|------|------|------|------|-------|
| Skåne | Helsingborg | 33,073 | 58,453 | 65,947 | 70,982 | 72,062 | 300,517 |
| | Kristianstad | 22,327 | 40,262 | 43,277 | 45,187 | 46,503 | 197,556 |
| | Lund | 46,565 | 58,808 | 60,314 | 60,882 | 60,200 | 286,769 |
| | Malmö | 45,518 | 67,018 | 68,385 | 66,515 | 62,332 | 309,768 |
| | Ystad | 12,504 | 23,071 | 24,153 | 25,243 | 26,005 | 110,976 |
| Stockholm | Danderyd | 0 | 0 | 63,468 | 84,052 | 85,511 | 233,031 |
| | Huddinge | 0 | 34,241 | 66,779 | 51,730 | 54,704 | 207,454 |
| | Solna | 0 | 27,039 | 46,065 | 17,747 | 13,163 | 104,014 |
| | Södersjukhuset | 0 | 0 | 8,241 | 99,009 | 94,369 | 201,619 |
| | Södertälje | 0 | 0 | 0 | 0 | 35,193 | 35,193 |
| Örebro | Örebro | 0 | 0 | 0 | 45,211 | 45,142 | 90,353 |
| Östergötland | Motala | 3 | 17,562 | 18,508 | 17,920 | 17,523 | 71,516 |
| | Linköping | 10 | 35,692 | 36,388 | 36,118 | 36,954 | 145,162 |
| | Norrköping | 3 | 34,843 | 36,718 | 37,713 | 37,187 | 146,464 |
| Total | | 160,003 | 396,989 | 538,243 | 658,309 | 686,848 | 2,440,392 |

**Table 3. Characteristics of patient visits by crowding category in primary analysis.**

| | | Crowding category | | | |
|---|---|---|---|---|---|
| | | 0%-75% | 75%-95% | 95%-100% | Total |
| **All visits, n** | | 1,830,378 | 488,021 | 121,993 | 2,440,392 |
| **Demographics, n (%)** | | | | | |
| Age | 18–39 | 544,691 (29.8) | 144,536 (29.6) | 36,975 (30.3) | 726,202 (29.8) |
| | 40–59 | 462,000 (25.2) | 123,680 (25.3) | 30,759 (25.2) | 616,439 (25.3) |
| | 60–79 | 541,085 (29.6) | 145,035 (29.7) | 36,452 (29.9) | 722,572 (29.6) |
| | 80 or older | 282,602 (15.4) | 74,770 (15.3) | 17,807 (14.6) | 375,179 (15.4) |
| Sex | Female | 947,709 (51.8) | 253,352 (51.9) | 61,918 (50.8) | 1,262,979 (51.8) |
| | Male | 882,669 (48.2) | 234,669 (48.1) | 60,075 (49.2) | 1,177,413 (48.2) |
| **Patient presentation at ED, n (%)** | | | | | |
| Arrival mode | Emergency medical services | 421,749 (23.0) | 111,057 (22.8) | 28,772 (23.6) | 561,578 (23.0) |
| | Walk-in or other | 1,340,745 (73.2) | 359,376 (73.6) | 88,340 (72.4) | 1,788,461 (73.3) |
| | Missing | 67,884 (3.7) | 17,588 (3.6) | 4,881 (4.0) | 90,353 (3.7) |
| Priority | 1—Red | 74,130 (4.0) | 19,852 (4.1) | 5,659 (4.6) | 99,641 (4.1) |
| | 2—Orange | 207,252 (11.3) | 56,051 (11.5) | 16,040 (13.1) | 279,343 (11.4) |
| | 3—Yellow | 589,668 (32.2) | 157,395 (32.3) | 38,561 (31.6) | 785,624 (32.2) |
| | 4—Green | 851,446 (46.5) | 224,974 (46.1) | 53,545 (43.9) | 1,129,965 (46.3) |
| | 5—Blue | 62,187 (3.4) | 17,174 (3.5) | 5,387 (4.4) | 84,748 (3.5) |
| | 6—Purple | 1,539 (0.1) | 392 (0.1) | 110 (0.1) | 2,041 (0.1) |
| | Missing | 44,156 (2.4) | 12,183 (2.5) | 2,691 (2.2) | 59,030 (2.4) |
| Chief complaint | Abdominal problem, GI bleeding | 15,905 (0.9) | 4,174 (0.9) | 1,050 (0.9) | 21,129 (0.9) |
| | Abdominal pain | 239,546 (13.1) | 63,462 (13.0) | 16,348 (13.4) | 319,356 (13.1) |
| | Abnormal lab test | 8,627 (0.5) | 2,184 (0.4) | 443 (0.4) | 11,254 (0.5) |
| | Arrythmia | 44,775 (2.4) | 11,788 (2.4) | 3,097 (2.5) | 59,660 (2.4) |
| | Back pain | 34,021 (1.9) | 9,140 (1.9) | 2,208 (1.8) | 45,369 (1.9) |
| | Cardiac arrest | 1,290 (0.1) | 326 (0.1) | 85 (0.1) | 1,701 (0.1) |
| | Chest or back injury | 15,468 (0.8) | 4,438 (0.9) | 1,116 (0.9) | 21,022 (0.9) |
| | Chest pain | 165,297 (9.0) | 45,390 (9.3) | 11,341 (9.3) | 222,028 (9.1) |
| | Decreased consciousness | 1,928 (0.1) | 546 (0.1) | 133 (0.1) | 2,607 (0.1) |
| | Diarrhea | 9,037 (0.5) | 2,462 (0.5) | 512 (0.4) | 12,011 (0.5) |
| | Dizziness | 50,117 (2.7) | 13,223 (2.7) | 3,020 (2.5) | 66,360 (2.7) |
| | Dyspnea | 114,678 (6.3) | 31,408 (6.4) | 8,400 (6.9) | 154,486 (6.3) |
| | Fever | 46,774 (2.6) | 12,398 (2.5) | 3,542 (2.9) | 62,714 (2.6) |
| | Head injury | 58,864 (3.2) | 15,725 (3.2) | 4,128 (3.4) | 78,717 (3.2) |
| | Local infection | 41,068 (2.2) | 11,024 (2.3) | 2,590 (2.1) | 54,682 (2.2) |
| | Lower extremity injury | 104,935 (5.7) | 28,426 (5.8) | 6,234 (5.1) | 139,595 (5.7) |
| | Multiple and/or major trauma | 9,577 (0.5) | 2,329 (0.5) | 927 (0.8) | 12,833 (0.5) |
| | Nausea, vomiting | 10,229 (0.6) | 2,943 (0.6) | 745 (0.6) | 13,917 (0.6) |
| | Neurological deficit, stroke | 57,367 (3.1) | 15,298 (3.1) | 3,565 (2.9) | 76,230 (3.1) |
| | Non-specific complaint | 121,642 (6.6) | 31,852 (6.5) | 8,372 (6.9) | 161,866 (6.6) |
| | Non-traumatic symptoms in extremity | 108,598 (5.9) | 28,585 (5.9) | 6,713 (5.5) | 143,896 (5.9) |
| | Other | 397,117 (21.7) | 104,736 (21.5) | 26,581 (21.8) | 528,434 (21.7) |
| | Seizures | 14,126 (0.8) | 3,589 (0.7) | 933 (0.8) | 18,648 (0.8) |
| | Syncope | 22,781 (1.2) | 5,760 (1.2) | 1,335 (1.1) | 29,876 (1.2) |
| | Upper extremity injury | 104,322 (5.7) | 28,373 (5.8) | 6,456 (5.3) | 139,151 (5.7) |
| | Urinary problems | 32,289 (1.8) | 8,442 (1.7) | 2,119 (1.7) | 42,850 (1.8) |
| **Timing of visit, n (%)** | | | | | |

*(Continued)*

**Table 3.** (Continued)

| | | Crowding category | | | |
|---|---|---|---|---|---|
| | | 0%-75% | 75%-95% | 95%-100% | Total |
| Shift | Day | 844,344 (46.1) | 224,598 (46.0) | 53,054 (43.5) | 1,121,996 (46.0) |
| | Evening | 742,474 (40.6) | 192,361 (39.4) | 29,448 (24.1) | 964,283 (39.5) |
| | Night | 243,560 (13.3) | 71,062 (14.6) | 39,491 (32.4) | 354,113 (14.5) |
| Weekend | Weekday | 1,198,634 (65.5) | 315,756 (64.7) | 64,740 (53.1) | 1,579,130 (64.7) |
| | Weekend or holiday | 631,744 (34.5) | 172,265 (35.3) | 57,253 (46.9) | 861,262 (35.3) |

proportion of visits arriving during night shifts and weekends were higher in the high crowding category.

## Main results

There were 41,737 deaths within 30 days of the ED visit. The total time at risk was 174,017 person-years and the average follow-up time was 26 days. The overall incidence rate was 24.0 deaths/100 person-years, with an incidence rate of 23.8 in the lowest reference category, 24.2 in the moderate category and 25.4 in the high crowding category (Table 4). The estimated adjusted HR (95% CI) was 1.02 (1.00–1.05) in the moderate crowding category with a p-value of 0.08 and 1.01 (0.96–1.05) in the high category. The estimated HRs for 7-day mortality, were slightly higher with HR of 1.05 (1.00–1.09) with a p-value of 0.04 in the moderate crowding category and 1.02 (0.94–1.10) in the high category (Table 5).

**Table 4. Association between crowding category and 30-day mortality.**

| | | Crowding category | | |
|---|---|---|---|---|
| Cohort | | 0%-75% | 75%-95% | 95%-100% |
| All hospitals | Number of deaths, n | 31,098 | 8,434 | 2,205 |
| | Person-years at risk, n | 130,547 | 34,789 | 8,681 |
| | Incidence rate, cases/100 person-years | 23.8 | 24.2 | 25.4 |
| | Adjusted[1] HR (95% CI) | Reference | 1.02 (1.00–1.05)[2] | 1.01 (0.96–1.05) |
| Skåne | Number of deaths, n | 16,480 | 4,509 | 1,167 |
| | Person-years at risk, n | 63,607 | 16,946 | 4,234 |
| | Incidence rate, cases/100 person-years | 25,9 | 26,6 | 27,6 |
| | Adjusted[1] HR (95% CI) | Reference | 1.03 (0.99–1.06) | 1.01 (0.95–1.07) |
| Stockholm | Number of deaths, n | 8,663 | 2,390 | 564 |
| | Person-years at risk, n | 41,367 | 11,024 | 2,745 |
| | Incidence rate, cases/100 person-years | 20.9 | 21.7 | 20.5 |
| | Adjusted[1] HR (95% CI) | Reference | 1.06 (1.01–1.11)[3] | 1.08 (0.98–1.18) |
| Östergötland | Number of deaths, n | 4,248 | 1,143 | 291 |
| | Person-years at risk, n | 19,667 | 5,243 | 1,315 |
| | Incidence rate, cases/100 person-years | 21.6 | 21.8 | 22.1 |
| | Adjusted[1] HR (95% CI) | Reference | 0.99 (0.93–1.06) | 0.97 (0.86–1.10) |

[1] stratified by hospital, adjusted for age, sex, priority, weekend, hour, arrival mode and chief complaint.

[2] P-value = 0.08 (non-significant).

[3] P-value = 0.02 (significant).

**Table 5. Association between crowding category and 7-day mortality.**

| Cohort | | Crowding category | | |
|---|---|---|---|---|
| | | 0%-75% | 75%-95% | 95%-100% |
| All hospitals | Number of deaths, n | 11,517 | 3,184 | 867 |
| | Person-years at risk, n | 16,867 | 4,488 | 1,113 |
| | Incidence rate, cases/100 person-years | 68.3 | 70.9 | 77.9 |
| | Adjusted[1] HR (95% CI) | Reference | 1.05 (1.00–1.09)[2] | 1.02 (0.94–1.10) |
| Skåne | Number of deaths, n | 6,462 | 1,776 | 492 |
| | Person-years at risk, n | 7,525 | 2,000 | 500 |
| | Incidence rate, cases/100 person-years | 85.9 | 88.8 | 98.4 |
| | Adjusted[1] HR (95% CI) | Reference | 1.04 (0.98–1.10) | 1.02 (0.92–1.13) |
| Stockholm | Number of deaths, n | 2,746 | 791 | 178 |
| | Person-years at risk, n | 5,890 | 1,576 | 380 |
| | Incidence rate, cases/100 person-years | 46.6 | 50.2 | 46.8 |
| | Adjusted[1] HR (95% CI) | Reference | 1.12 (1.03–1.22)[3] | 1.11 (0.94–1.32) |
| Östergötland | Number of deaths, n | 1,686 | 460 | 124 |
| | Person-years at risk, n | 2,738 | 727 | 189 |
| | Incidence rate, cases/100 person-years | 61.6 | 63.3 | 65.6 |
| | Adjusted[1] HR (95% CI) | Reference | 1.00 (0.89–1.12) | 1.00 (0.82–1.22) |

[1] stratified by hospital, adjusted for age, sex, priority, weekend, hour, arrival mode and chief complaint.

[2] P-value = 0.04 (significant).

[3] P-value = 0.01 (significant).

## Subgroup analysis

The Stockholm county had the highest ED LOS with a mean of 320 min compared to 275 min in Skåne and 216 min in Östergötland. The median age was similar with median 55 years in Stockholm and 56 years in Skåne and Östergötland. The EMS arrival proportion was highest in Stockholm county with 30% compared to 18% in Skåne and 29% in Östergötland. The proportion of patients admitted to inpatient care were 37% in Stockholm while it was 29% in Skåne and 23% in Östergötland. In the subgroup analysis of all patients we found no statistically significant association between crowding and mortality in Skåne and Östergötland counties. The point estimates for the HR's in Skåne county were slightly elevated but not statistically significant. In the Stockholm county the estimated adjusted HR for 30-day mortality was 1.06 (1.01–1.11) in the moderate crowding category, and 1.08 (0.98–1.18) in the high category (Table 4). The subgroup analysis for admitted patients showed similar but slightly higher HR estimates with statistically significant results in the moderate category for Skåne with HR 1.04(1.00–1.08) and statistically significant results for Stockholm in both categories with HR of 1.06 (1.01–1.11) in the moderate crowding category and 1.11 (1.01–1.22) in the high category (Table 6). The number of included visits in the Stockholm subgroup analysis were 759,838 for all patients and 284,275 for admitted patients.

## Sensitivity analysis

The sensitivity analysis including all hospitals with a 30-day follow-up period but not including triage priority in the regression model showed similar results as the primary model with an estimated adjusted HR of 1.02 (1.00–1.05) with a p-value of 0.06 (non-significant) in the moderate crowding category, and 1.01 (0.97–1.06) in the high category.

**Table 6. Association between crowding category and 30-day mortality for admitted patients.**

| Cohort | | Crowding category | | |
| --- | --- | --- | --- | --- |
| | | 0%-75% | 75%-95% | 95%-100% |
| All hospitals | Number of deaths, n | 26,365 | 7,293 | 1,919 |
| | Person-years at risk, n | 39,516 | 10,532 | 2,628 |
| | Incidence rate, cases/100 person-years | 66.7 | 69.2 | 73.0 |
| | Adjusted[1] HR (95% CI) | Reference | 1.04 (1.01–1.07) | 1.03 (0.98–1.08) |
| Skåne | Number of deaths, n | 14,058 | 3,953 | 986 |
| | Person-years at risk, n | 18,101 | 4,828 | 1,206 |
| | Incidence rate, cases/100 person-years | 77.7 | 81.9 | 81.7 |
| | Adjusted[1] HR (95% CI) | Reference | 1.04 (1.00–1.08)[2] | 1.01 (0.94–1.08) |
| Stockholm | Number of deaths, n | 7,615 | 2,087 | 536 |
| | Person-years at risk, n | 15,231 | 4,062 | 1,016 |
| | Incidence rate, cases/100 person-years | 50.0 | 51.4 | 52.8 |
| | Adjusted[1] HR (95% CI) | Reference | 1.06 (1.01–1.12) | 1.11 (1.01–1.22) |
| Östergötland | Number of deaths, n | 3,201 | 888 | 221 |
| | Person-years at risk, n | 4,291 | 1,140 | 288 |
| | Incidence rate, cases/100 person-years | 74.6 | 77.9 | 76.7 |
| | Adjusted[1] HR (95% CI) | Reference | 1.03 (0.96–1.12) | 1.02 (0.88–1.17) |

[1] stratified by hospital, adjusted for age, sex, priority, weekend, hour, arrival mode and chief complaint.

[2] P-value = 0.03 (significant).

## Limitations

The study is based on data from the SVAR registry, that receive information from the EHR of each participating ED. The registry strives to use the same definitions of the variables at all the sites. However, there may be differences in how data is defined, registered and uploaded to the system across the group of included hospitals. Through quality control and logical testing, we have found and corrected some minor irregularities in the registry data. One example was the inconsistent matching logic of temporary personal ID's for 2% of the visits so these visits were removed. Another was that LOS information was missing for 1% of the visits, so we replaced them with the mean LOS to enable a calculation of the ED census including all visits. Furthermore, priority was defined differently in the Stockholm county. To counter this, we added a sensitivity analysis not including triage priority in the statistical model, and it showed similar results. Altogether, this together with other unknown data issues may have introduced bias. The SVAR registry is growing and some of the EDs were not included from the start. This means that the relative share of visits between the EDs changed over time. The statistical model was stratified by ED and used calendar date as underlying time dimension, so it was able to manage this variation in coverage over time together with other known or unknown seasonality effects. The stratification of the model by ED allows the baseline risk to vary between the sites. However, it assumes that the estimated hazard ratios are the same across EDs. This may have reduced the accuracy of the model since the EDs here are more heterogenous compared to our prior study [10] with a similar methodology. The definition of the proxy-measure for crowding is new and has only been tested in one prior study [10] and by the same research group. It therefore needs further validation. The exposure is defined as the actual census divided with the predicted census. Assuming that the variation of the actual census is constant, the exposure variation will be higher when the predicted census (denominator) is small. Indeed, during nights and weekends where the predicted census was lower, the share

of visits in the high crowding category increased. Arrival times are thus associated with the exposure and can also be linked to the outcome, as the case-mix probably varies with the timing of arrival [20,21]. In addition, there could also be a potential "weekend effect" [20,21] where the outcome is worse outside of normal office hours. Arrival time and weekday/weekend are therefore important to include in the statistical model together with age, priority and other potential case-mix factors. Still, due to the study design there may be residual confounding that we have not accounted for. Performing a subgroup analysis for admitted patients may be problematic as we risk introducing a source of confounding-by-severity. In cases when the reason for crowding is a lack of inpatient beds in the hospital, there is a risk that the threshold for admission increases, which could imply that the group of admitted patients are sicker in these instances. Even if we have adjusted for important patient characteristics like age, sex, arrival mode, triage priority and chief complaint, we increase the risk of residual confounding in this subgroup analysis.

## Discussion

Leveraging the national quality registry SVAR, we analysed 2,440,392 visits from 1,142,631 unique patients from 14 EDs and four Swedish counties during the years 2015–2019. In the main analysis including all EDs, we found mixed results. Crowding was significantly associated with 7-day mortality, but not with 30-day mortality. The mixed and weak results in the overall analysis are likely due to differences between the counties that diluted the results in the main analysis. When three of the counties were analysed separately, and when only including admitted patients, we found clear evidence of an association with 30-day mortality in the Stockholm county, mixed results in Skåne, but no signs of an association in Östergötland. The estimated risk for admitted patients in the Stockholm county was 6% higher in the moderate crowding category and 11% higher in the high category compared to the reference. In our prior study [10], in the Stockholm county in 2012–2016 (which included two more EDs), the estimated HR was 1.08 (1.03–1.14) in the high crowding category which is consistent with the present results. Compared to our prior study, a new finding is the suggested mortality association already in the moderate crowding category. We lack sufficient knowledge, but key crowding indicators like waiting time for physician and LOS have steadily increased during the period 2012 to 2019 [22,23], indicating increased absolute levels of ED crowding. In the Skåne county the findings were mixed, and in Östergötland we found no signs of an association between crowding and mortality. This is hopeful as it shows that the association between ED crowding and increased mortality is not universal and potentially could be avoided in line with the results that was found in a Belgian academic teaching hospital [6] and an inner-city hospital in the Netherlands [24].

In our prior study [10], the relative risk was translated to an absolute risk (95% CI) of 6 (2–9) deaths per 100,000 ED visits. In a similar analysis for the present study, based on the results for admitted patients, 23 (3–42) deaths per 100,000 visits would occur in Stockholm, which is a substantial number of potentially avoidable deaths. We do not know why the association between ED crowding and increased mortality was mainly found in the Stockholm county. The Stockholm EDs tend to be larger in terms of annual patient volumes and have longer average ED LOS. The size of the ED is known to impact crowding. [14,15] and a long average LOS indicate a higher absolute level of crowding. Both the share of patients arriving with ambulance or helicopter and admitted to inpatient care was higher in Stockholm, suggesting that the average patient was likely sicker in Stockholm. This is probably at least partly due to the recent introduction of co-located urgent care centers with primarily general practitioner physicians who take care of lower priority patients. Another important difference between the

counties is the hospital bed occupancy rate. According to a national statistics database [25], the average hospital bed occupancy weighted with the visit volumes included in the study was 101% in Stockholm, 92% in Skåne and 81% in Östergötland. In an earlier study we found that a high hospital bed occupancy is closely linked to an increased ED workload with longer LOS and fewer admissions to inpatient care suggesting tougher prioritizations [26]. It is possible that the lower bed occupancy level in Skåne and Östergötland functions as a buffer, limiting the most dangerous consequences of ED crowding. Recent findings from France [27] and New Zeeland [28] also indicate that boarding of admitted patients is associated with increased mortality and that the output [29] dimension and access to inpatient beds is critical in the association between crowding and mortality.

In summary, the results for the association between our ED crowding measure and increased mortality were mixed and varied by county. In one county there were statistically significant associations in line with prior findings [10–13], while there were mixed or no associations in the other counties. Since the association does not seem to be universal, it may be avoidable. Factors that influence the association between crowding and mortality at different EDs are still unknown but a high hospital bed occupancy, impacting admitted patients may play a role.

## Author Contributions

**Conceptualization:** Björn af Ugglas, Therese Djärv, Martin J. Holzmann.

**Data curation:** Björn af Ugglas, Per Lindmarker, Ulf Ekelund.

**Formal analysis:** Björn af Ugglas.

**Investigation:** Björn af Ugglas.

**Methodology:** Björn af Ugglas, Per Lindmarker, Ulf Ekelund, Martin J. Holzmann.

**Project administration:** Björn af Ugglas.

**Resources:** Björn af Ugglas, Therese Djärv, Martin J. Holzmann.

**Software:** Björn af Ugglas.

**Supervision:** Björn af Ugglas, Therese Djärv, Martin J. Holzmann.

**Validation:** Björn af Ugglas.

**Visualization:** Björn af Ugglas.

**Writing – original draft:** Björn af Ugglas.

**Writing – review & editing:** Björn af Ugglas, Per Lindmarker, Ulf Ekelund, Therese Djärv, Martin J. Holzmann.

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
