## [Decision Letter · Decision Letter 0]

6 Jan 2021

PONE-D-20-36959

Emergency department crowding and mortality in 14 Swedish emergency departments, a cohort study leveraging the Swedish emergency registry (SVAR)

PLOS ONE

Dear Dr. af Ugglas,

Thank you for submitting your manuscript to PLOS ONE. After careful consideration, we feel that it has merit but does not fully meet PLOS ONE’s publication criteria as it currently stands. It is a well-written paper that covers an important topic. The study is well designed and conducted. However, the reviewers highlight some aspects that need to be answered. Therefore, we invite you to submit a revised version of the manuscript that addresses the points raised during the review process.

We look forward to receiving your revised manuscript.

Kind regards,

Juan F. Orueta, MD, PhD

Academic Editor

PLOS ONE

Journal Requirements:

"Dr. Djärv was supported by the Stockholm County Council (clinical research appointment).

Dr. Holzmann reports receiving consultancy honoraria from Actelion, Idorsia, and Pfizer. He holds research positions funded by the Swedish Heart- Lung Foundation (grant 20170804) and the ALF agreement between the Stockholm County Council and Karolinska Institutet (grant 20170686)."

**Comments to the Author**

1. Is the manuscript technically sound, and do the data support the conclusions?

Reviewer #1: Yes

Reviewer #2: Yes

2. Has the statistical analysis been performed appropriately and rigorously? 

Reviewer #1: Yes

Reviewer #2: Yes

3. Have the authors made all data underlying the findings in their manuscript fully available?

Reviewer #1: Yes

Reviewer #2: Yes

4. Is the manuscript presented in an intelligible fashion and written in standard English?

Reviewer #1: Yes

Reviewer #2: Yes

5. Review Comments to the Author

Reviewer #1: Thank you for a clear and well structured paper. My only difficult was trying to understand the "expected census" despite your very best efforts. I wonder if it would be helpful to give a "for instance".

Reviewer #2: Thank you for the opportunity to review this important and interesting study. The study is conducted in Sweden over a four year time window and uses administrative data drawn from a national quality improvement registry and linked to population vital statistics to explore the relationship between Emergency Department crowding and mortality. Drawing from over 2 400,000 visits and over 1,100,000 unique patients the study establishes definitions for the crowding exposure and using statistical methods to adjust for important covariates determines measures of association between seven day and 30 day mortality in this large robust database. Of note this is a follow-up study of a previous project conducted in Sweden where the novel crowding measure of observed over expected census was first developed though not clearly validated from the information provided in the paper. Interestingly this first report noted an association between crowding and 30 day mortality, a finding not reproduced with this report.

The findings are somewhat mixed in that the association between crowding and mortality only achieve statistical significance in one of the four counties under consideration. This is noted in the Stockholm County where only the moderate degree of crowding was sufficiently powered to yield a robust relationship. Adding further complexity to this scenario the Stockholm County uses different criteria to code the urgency of the Emergency Department presentation. Of note this discrepancy is addressed in a sensitivity analysis which did not substantially indicate a change in the conclusions.

An additional finding not well explained is how there appears to be a relationship between seven-day mortality in that county but not with the 30-day primary outcome.

While overall I believe this is a well conducted and nicely reported study, I do have concerns however. The primary issue I feel is the lumping of both admitted and unadmitted patients in the analysis. The reason for this is that one could argue that the impact of crowding on Emergency Department care would manifest mostly on the pressure to discharge potentially sick patients due to high hospital occupancy whereas mortality occurring for admitted patients maybe less directly linked to the quality of care provided in the Emergency Department itself and therefore the impact of crowding. Therefore I would favor a subgroup analysis based on this.

I also think there needs to be better justification for drawing from the most common 25 presenting complaints for this analysis. Could we not imagine a link to time sensitive conditions where treatment delays may have a greater impact on outcomes as reported in this paper? https://pubmed.ncbi.nlm.nih.gov/31640561/

6. PLOS authors have the option to publish the peer review history of their article (what does this mean?). If published, this will include your full peer review and any attached files.

Reviewer #1: No

Reviewer #2: **Yes: **Eddy Lang

---

## [Author Response · Author response to Decision Letter 0]

21 Jan 2021

Please see attached document "Response to reviewers"

---

## [Decision Letter · Decision Letter 1]

16 Feb 2021

Emergency department crowding and mortality in 14 Swedish emergency departments, a cohort study leveraging the Swedish emergency registry (SVAR)

PONE-D-20-36959R1

Dear Dr. af Ugglas,

We’re pleased to inform you that your manuscript has been judged scientifically suitable for publication and will be formally accepted for publication once it meets all outstanding technical requirements.

Kind regards,

Juan F. Orueta, MD, PhD

Academic Editor

PLOS ONE

Reviewers' comments:

Reviewer's Responses to Questions

**Comments to the Author**

1. If the authors have adequately addressed your comments raised in a previous round of review and you feel that this manuscript is now acceptable for publication, you may indicate that here to bypass the “Comments to the Author” section, enter your conflict of interest statement in the “Confidential to Editor” section, and submit your "Accept" recommendation.

Reviewer #1: All comments have been addressed

Reviewer #2: All comments have been addressed

2. Is the manuscript technically sound, and do the data support the conclusions?

Reviewer #1: Yes

Reviewer #2: Yes

3. Has the statistical analysis been performed appropriately and rigorously? 

Reviewer #1: Yes

Reviewer #2: Yes

4. Have the authors made all data underlying the findings in their manuscript fully available?

Reviewer #1: Yes

Reviewer #2: Yes

5. Is the manuscript presented in an intelligible fashion and written in standard English?

Reviewer #1: Yes

Reviewer #2: Yes

6. Review Comments to the Author

Reviewer #1: This article uses a surrogate measure of ED crowding to demonstrate an association between crowding and 7day mortality. Thank you for addressing the previous comments. I have no further suggested improvements.

Reviewer #2: Nice revision - key issues have been addressed. There is far more clarity related to data sources and the analysis undertaken. Messaging is clearer as well. The response to reviewers section is well-presented and rich in detail as well as appropriate responses.

7. PLOS authors have the option to publish the peer review history of their article (what does this mean?). If published, this will include your full peer review and any attached files.

Reviewer #1: No

Reviewer #2: No

---

## [Editor Report · Acceptance letter]

18 Feb 2021

PONE-D-20-36959R1 

Emergency department crowding and mortality in 14 Swedish emergency departments, a cohort study leveraging the Swedish emergency registry (SVAR) 

Dear Dr. af Ugglas:

I'm pleased to inform you that your manuscript has been deemed suitable for publication in PLOS ONE. Congratulations! Your manuscript is now with our production department. 

Kind regards, 

on behalf of

Dr. Juan F. Orueta 

Academic Editor

PLOS ONE